# A silicon transporter gene required for healthy growth of rice on land

Namiki Mitani-Ueno [1,4], Naoki Yamaji[1,4], Sheng Huang [1,4], Yuma Yoshioka [2], Takaaki Miyaji [2,3] & Jian Feng Ma [1] ✉

Silicon (Si) is the most abundant mineral element in the earth's crust. Some plants actively accumulate Si as amorphous silica (phytoliths), which can protect plants from stresses. Here, we report a gene (*SIET4*) that is required for the proper accumulation and cell-specific deposition of Si in rice and show that it is essential for normal growth. *SIET4* is constitutively expressed in leaves and encodes a Si transporter. SIET4 polarly localizes at the distal side of epidermal cells and cells surrounding the bulliform cells (motor cells) of the leaf blade, where Si is deposited. Knockout of *SIET4* leads to the death of rice in the presence but not absence of Si. Further analysis shows that *SIET4* knockout induces abnormal Si deposition in mesophyll cells and the induction of hundreds of genes related to various stress responses. These results indicate that SIET4 is required for the proper export of Si from leaf cells to the leaf surface and for the healthy growth of rice on land.

Silicon (Si) is the most abundant mineral element in the soil from which the plant roots take up the mineral. The extent of Si accumulation varies with the plant species, ranging from 0.1% to 10% on a dry weight basis. Si accumulation is high in primitive land plants including *Bryophyta*, *Lycopsida* and *Equisetopsida* of *Pteridophyta*[1,2]. *Polypodiophyta* have both Si-accumulating and non-accumulating species while there are no high Si-accumulating species in *Gymnospermae*. In *Angiospermae*, although species belonging to *Gramineae* and *Cyperaceae* show high Si accumulation, most plant species do not[1,2]. The benefit of high-Si accumulation is protection from various stresses including abiotic and biotic stresses[3–5]. A typical example is rice (*Oryza sativa*), which accumulates up to 10% of Si in the shoots on a dry weight basis[6]. This high accumulation of Si is required for stable and high grain yield of rice[7], because Si deposition mitigates the damages caused by pathogens, insect pests, drought, salt, metal toxicity, lodging, and nutrient imbalance stresses[1]. Si in soil solution is present in the form of silicic acid {Si(OH)$_4$}, a non-charged molecule, which is taken up by the plant roots. Transporters involved in Si uptake have been identified in rice and other plant species[8–20]. Si uptake in rice is mediated by two different types of transporters (Lsi1 and Lsi2), which function as influx

and efflux transporters of Si, respectively. Both Lsi1 and Lsi2 are localized at the exodermis and endodermis in the roots, but show different polar localization. Lsi1 is localized at the distal side and Lsi2 is localized at the proximal side[8,9]. Si is first imported into the symplast by Lsi1 at the distal side of the exodermal cells and then exported by Lsi2 at the proximal side to the apoplastic connections. Si is further imported into the symplast of the endodermis by Lsi1 localized on the distal side of the endodermis, and is exported to the stele by Lsi2 localized on the proximal side of the endodermis[8,9]. Analysis of these transporters showed that Si accumulation is closely associated with the expression level, localization and polarity of these transporters in different plant species[21,22]. After uptake, more than 95% of Si is rapidly released to the xylem by both Lsi2 and Lsi3 in rice. Lsi3 is a homolog of Lsi2, and is localized at the root pericycle cells[23]. Si in the xylem sap is also transiently present in the form of monosilicic acid[24]. Unloading of this Si from the xylem is mediated by Lsi6, a homolog of Lsi1[25]. Lsi6 is polarly localized at the adaxial side of the xylem parenchyma cells in the leaf sheaths and leaf blades[25]. With water loss due to transpiration, silicic acid is gradually concentrated and polymerized to amorphous silica, forming silica cells and silica bodies or silica bulliform cells

[1]Institute of Plant Science and Resources, Okayama University, Chuo 2-20-1, Kurashiki 710-0046, Japan. [2]Graduate School of Medicine, Dentistry and Pharmaceutical Sciences, Okayama University, Tsushima Naka 1-1-1, Kita Okayama 700-8530, Japan. [3]Department of Genomics & Proteomics, Advanced Science Research Center, Okayama University, Tsushima Naka 1-1-1, Kita Okayama 700-8530, Japan. [4]These authors contributed equally: Namiki Mitani-Ueno, Naoki Yamaji, Sheng Huang. ✉e-mail: maj@rib.okayama-u.ac.jp

(motor cells) in specific tissues and cells [1,26]. Silica cells are located on the leaf epidermis along the vascular bundles, in a dumbbell shape, while silica bodies are silicified bulliform cells of rice leaves, known as phytoliths or plant opal. Since the deposited silica takes the cell shape of certain plant species, these phytoliths have been used in plant taxonomy and as markers in archeological and paleoecological researches where they are used to determine past agricultural practices, human diet and environment [27]. In addition, Si is also deposited beneath the cuticle, forming cuticle–silica double layers [1]. The molecular mechanisms underlying the cell-specific deposition of Si have not been identified. In the present study, we identified a transporter, SIET4 (Silicon Efflux Transporter 4), which enables healthy growth of rice on land by controlling specific Si deposition in the leaves.

## Results

### Knockout of *SIET4* results in death of rice grown in soil

There are four homologs of Lsi2, also known as Silicon Efflux Transporter (SIET)[9]. In the present study, we functionally characterized one of them, SIET4. SIET4 shares 49% identity with Lsi2. To understand the role of this gene, we generated the knockout lines by the CRISPR/Cas9 technique. We used two mutants; *siet4-1*, and *siet4-2*, which had 1-bp insertion at the first and second exon, respectively, for further analysis (Supplementary Fig. 1). When the mutants were grown in soil with their wild-type rice (WT), surprisingly and unexpectedly, the growth of the two mutants was significantly inhibited compared with the WT and finally led to death (Fig. 1a, b).

WT and the two mutants were grown in a nutrient solution with or without Si supplied until maturation to investigate whether this growth inhibition was caused by Si in the solution. WT and two mutants showed similar growth when Si was not supplied (Fig. 1c). However, in the solution with Si supplied, the growth of both the roots and shoots was severely inhibited in the mutants (Fig. 1d); the dry weight of the roots and shoots in the mutants was less than 10% of that in the WT (Supplementary Fig. 2a, b). The mutants died without setting seed (Fig. 1 and Supplementary Fig. 2c). Addition of Si to the solution enhanced the growth of WT (Fig. 1c, d and Supplementary Fig. 2a).

We further compared the time-dependent growth of WT and mutants grown with and without 1 mM Si in the solution. In the solution without Si supply, the mutants and WT showed similar growth for up to 45 days (Supplementary Fig. 3a). However, in the solution with Si, the growth of the mutants was gradually poor compared with WT (Supplementary Fig. 3b). At day 45 in the solution with Si supplied, the fresh weight of mutants was only 22% of WT. Various symptoms were observed in the leaves of the mutants grown with Si in the solution. For example, after a few days in the solution with Si, the mutants developed leaves with a small white spot on the surface, became yellow in color, or twisted, but the WT showed no such changes (Supplementary Fig. 4). The SPAD value of the newest fully expanded leaf blade was decreased by 25% in the mutants compared with WT when grown in the solution with Si (Supplementary Fig. 5), whereas no difference was found between WT and mutants grown without Si supplied.

We also compared the growth of WT and mutants in a Si dose-responsive manner. Since rice has a high ability to take up Si, Si in the solution will be immediately depleted at a low Si concentration. To maintain a relatively stable level of Si in the solution, we grew WT and the *siet4* mutants in a pot filled with river sand containing a very small amount of soluble Si and added different rates of Water Silica (a slow-releasing pure Si fertilizer) to each pot, to obtain different Si concentrations in the solution (Supplementary Fig. 6a). These Si concentrations were comparable to those found in the natural soil solution of different soils. With increasing Water Silica application rates, the growth of WT was enhanced, but that of *siet4-1* mutants was progressively inhibited (Supplementary Figs. 6b, 7).

### SIET4 is involved in Si deposition, not in Si uptake

To understand the mechanism underlying Si-induced growth inhibition in the mutants, we determined the Si accumulation in WT and the mutants. The shoot Si concentration in the mutants was similar to that in WT grown in both soil and hydroponic solution with Si supplied (Fig. 2a, b). The Si concentration in the shoots was below the detectable level in both WT and mutants grown in solution without Si supply. A short-term uptake experiment also showed no difference in Si uptake between WT and mutants (Supplementary Fig. 8a). The Si concentration in the xylem sap of different lines was similar (Supplementary Fig. 8b). In addition, the concentration of germanium (Ge), an analog of Si, in the roots and shoots of plants grown in a solution with Ge supplied in the mutants was similar to that in WT (Supplementary Fig. 9). There was no consistent difference in the concentration of other mineral elements in the roots and shoots of different lines (Supplementary Fig. 9). These results indicate that SIET4 is not involved in Si uptake.

By contrast, the pattern of Si deposition in WT was different from that in the mutants. Comparison of the Si deposition pattern detected by the Laser Ablation Inductively Coupled Plasma Mass Spectrometry (LA-ICP-MS) technique showed that Si was mainly accumulated in the apoplastic area (metabolism-inactive sites) of the leaf surface and at the bulliform cells (motor cells), but not in the mesophyll cells in WT (Fig. 2c). However, in the two *siet4* mutants, Si accumulation was decreased in the leaf surface, and abnormal deposition was observed in the mesophyll cells (Fig. 2c). There was no difference in the accumulation pattern of P and Mn as controls between the WT and mutants (Supplementary Fig. 10). Furthermore, the amount of Si deposition in the leaf surface measured by SEM-EDX after 24 h exposure to the solution containing Si was significantly lower in the mutants than in WT (Fig. 2d). These results indicate that SIET4 is involved in tissue-specific Si deposition in the leaves.

### Transport activity of SIET4 protein

To understand the function of SIET4, we tested its transport activity, expression pattern and tissue/cellular localization. First we tested the transport activity of SIET4 for Si using the proteoliposome method [28]. We used Lsi2 as a positive control, which is an efflux transporter for Si[9]. The purified protein fraction exhibited a major protein band with an expected apparent molecular mass of SIET4 or Lsi2 (Fig. 3a). These proteins were further confirmed by western blot analysis using the anti-6×His antibody (Fig. 3a). Since the protein is bi-directionally reconstituted into liposomes in this assay system, this in vitro method allows determination of the influx or efflux transport activity by changing the driving force such as the pH inside and outside the liposomes [28]. Similar to Lsi2, the proteoliposomes containing purified recombinant SIET4 protein showed transport activity for Si when there was a pH gradient {outside pH 7.5 (similar to cytosol) versus inside pH 6.0 (similar to apoplast)} compared with the control (liposome without protein) (Fig. 3b). This indicates that SIET4 is an efflux transporter in plants. This transport activity disappeared when the pH was the same inside and outside of the proteoliposomes (Fig. 3b). The transport activity was almost abolished in the presence of carbonyl cyanide *m*-chlorophenylhydrazone (CCCP), a proton ionophore (Fig. 3b). Addition of 2 mM Ge significantly decreased the Si transport activity by SIET4 and Lsi2 (Fig. 3b).

SIET4-mediated transport for Si at various concentrations and at external pH 7.5 and pH 6.0 was characterized along with time. SIET4-mediated Si transport was much higher at pH 7.5 than at pH 6.0 (Supplementary Fig. 11a). The activity increased with time, and was saturated at 2 min (Supplementary Fig. 11a). The activity increased with increasing Si concentrations in the external solution (Supplementary Fig. 11b), while the activity was

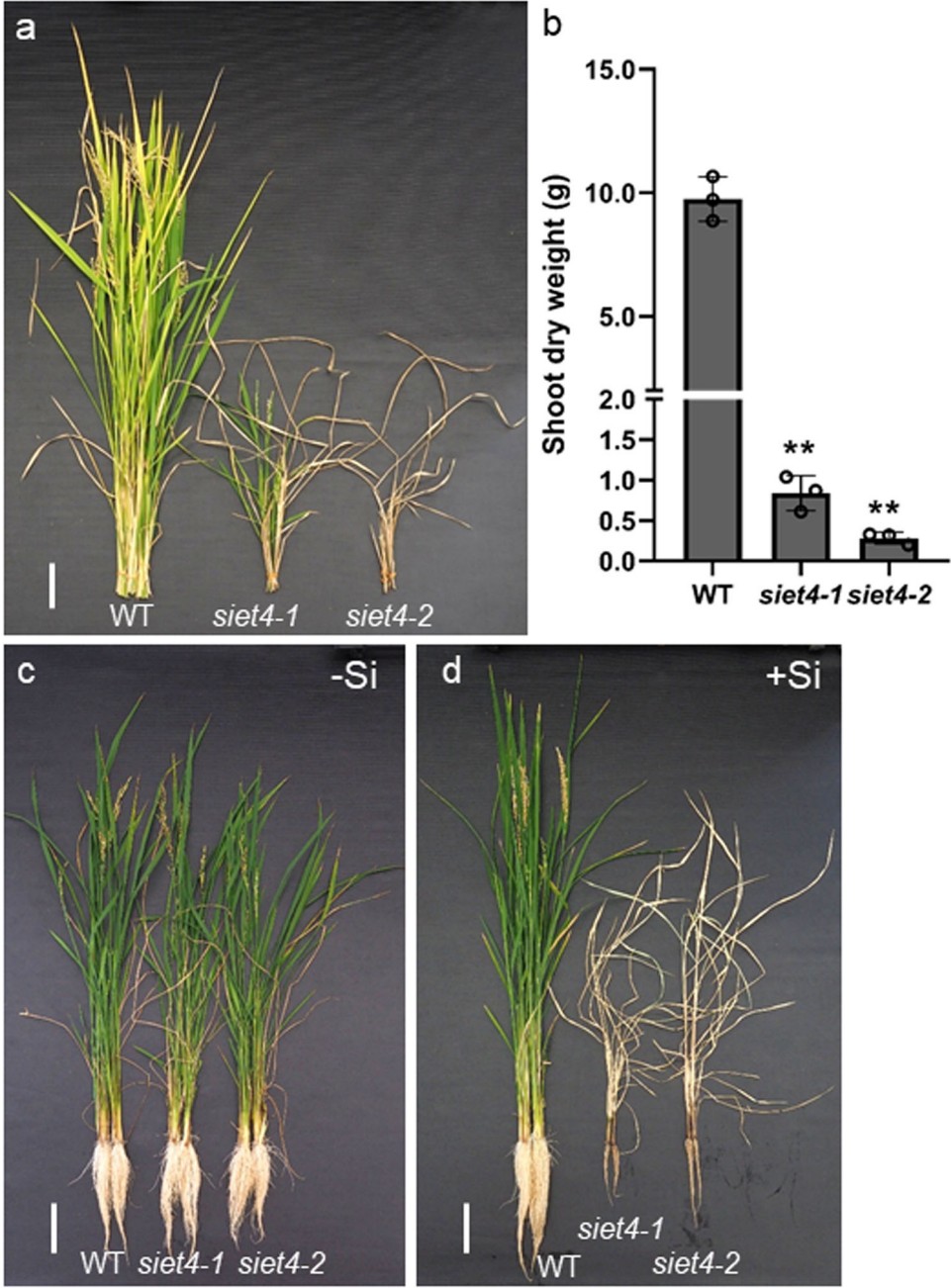

**Fig. 1 | Knockout of *SIET4* led to death of rice in the presence of Si.** Phenotypes (**a**) and shoot dry weight (**b**) of wild-type rice (WT) and two *siet4* mutants grown in soil. The WT and two mutants were grown in a pot soil until maturation in a greenhouse. Photo was taken at harvest. Data are means ± SD (*n* = 3 biologically independent plants). ** indicates significant difference compared with WT (*P* < 0.01). Phenotypes of WT and *siet4* mutants grown in a hydroponic solution without (**c**) or with Si (**d**). The plants of both WT and mutants were grown in a nutrient solution containing 0 or 1 mM Si as silicic acid until maturation. Scale bars for (**a, c, d**) = 10 cm.

higher at pH 7.5 that at pH 6.0 irrespective of Si concentration. These results indicate that like Lsi2, SIET4 is a Si efflux transporter driving by proton gradient.

## Expression profiles of *SIET4*

We then investigated the expression pattern of *SIET4* at various organs and stages. *SIET4* was expressed in all organs tested throughout the whole growth period (Fig. 4a), but showed higher expression in the leaf sheath and leaf blade at the vegetative growth stage. At the reproductive stage, higher expression was also found in nodes, rachis, peduncle, spikelet, and husk, where Si accumulation occurs (Fig. 4a). Further analysis showed that the leaf sheath and leaf blade showed similar expression levels, but the old leaves tended to show higher expression than the new leaves (Fig. 4b).

The response of *SIET4* expression to Si supply was also examined. The expression of *SIET4* in the shoot was unaffected by Si supply (Fig. 4c), indicating that *SIET4* is constitutively expressed in the shoots.

## Tissue and cellular localization of SIET4

We investigated the tissue-specificity of the localization of SIET4 by immunostaining using a specific antibody against SIET4. In the leaf blade of young leaves, the signal was observed only in the epidermal cells (Fig. 5a, b). Si supply did not affect the localization of SIET4 (Fig. 5a, b). In the leaf blade and sheath of the mature leaf with or

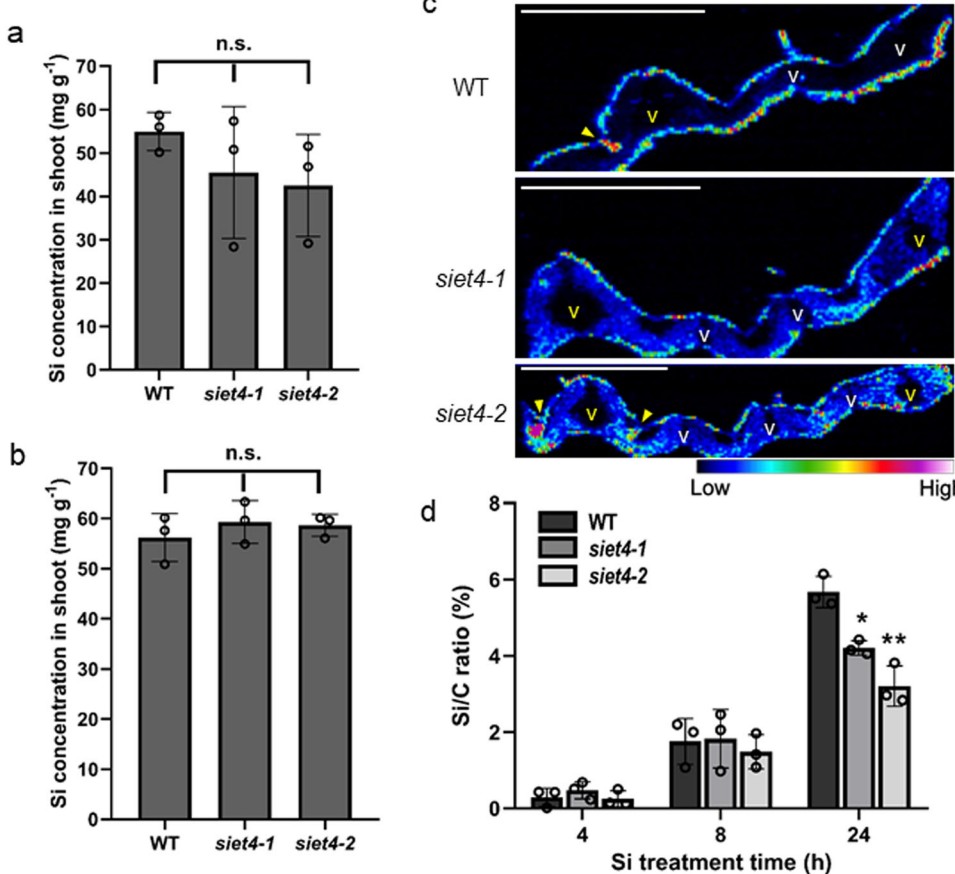

**Fig. 2 | Accumulation and deposition pattern of Si in rice leaves.** Si concentration in the shoot of wild-type rice (WT) and two *SIET4* knockout mutants (*siet4-1*, *siet4-2*) grown in soil (**a**) or hydroponic solution with Si (**b**). WT and two mutants were grown in a pot soil or a nutrient solution containing 1 mM Si until maturation in a greenhouse. Si concentration was determined by the colorimetric method after digestion with $HNO_3$-$H_2O_2$-HF. Data are means ± SD ($n$ = 3 biologically independent plants). Statistical analysis was performed by ANOVA followed by Tukey's test. n.s., not significant ($P > 0.05$). **c** Deposition pattern of Si detected by LA-ICP-MS. Seedlings (27-day-old) were put in a nutrient solution containing 1 mM Si. After 6 days, the leaf blade of the youngest fully expanded leaf was sampled for preparing cross-sections, followed by subjecting to LA-ICP-MS analysis. Yellow and white v show large and small vascular bundles, respectively. Yellow arrow heads indicate silicified bulliform cells (sbc). **d** Time-dependent deposition of Si in rice leaf surface. Seedlings (36-day-old) were put in a nutrient solution containing 1 mM Si. At the indicated time points, the central area of the youngest fully expanded leaf blade was examined for Si on the leaf by SEM-EDX with surface observation mode (5 kV acceleration voltage). Data are means ± SD ($n$ = 3 biologically independent leaves). * and ** indicate significant difference compared with WT ($P < 0.05$ or $<0.01$).

without Si supplied, the signal was also observed in the epidermal cells (Fig. 5d, e, g, h). Close observation revealed that SIET4 was also expressed in neighboring cells of bulliform cells (motor cells) and showed polar localization facing the major bulliform cell in leaf blade (Fig. 5j, k). SIET4 also showed polar localization mainly at the distal side in the epidermal cells of the leaf blade and leaf sheath (Fig. 5j, l). No signal was observed in the leaf blade and leaf sheath of *SIET4* knockout line (Fig. 5c, f, i), indicating the specificity of the antibody used for SIET4.

To examine the subcellular localization of SIET4, we performed double staining of SIET4 and HDEL as an ER marker together with DAPI for nuclear localization. The result showed that SIET4 was mainly localized at the plasma membrane of epidermal cells (Supplementary Fig. 12), but partially overlapped with HDEL.

### Transcriptomic analysis of *siet4* mutants

To understand the mechanism underlying the Si-induced plant death in *siet4* mutants (Fig. 1), we compared transcript changes of genes in the leaf blades between WT and *siet4-1* mutant with and without Si supply for 24 h by RNA-seq. By comparing plants with and without Si supplied, 297 genes and 251 genes were up-regulated and down-regulated by Si (≥2-fold), respectively, in the WT, whereas 1250 genes and 793 genes were up-regulated and down-regulated, respectively, by

Si in the mutant (Supplementary Fig. 13a). Among the up-regulated and down-regulated genes, 165 and 75 genes, respectively, overlapped. To understand Si-induced toxicity in the mutants, we further performed gene ontology (GO) analysis of the genes up-regulated only in the mutant. The results showed that these genes were associated with the GO terms stress response such as response to chitin (GO:0010200), regulation of response to osmotic stress (GO:0047484), response to wounding (GO:0009611), response to fungus (GO:0009620), regulation of response to stresses (GO:0080134), etc. (Supplementary Fig. 13b). On the other hand, the genes down-regulated only in the mutant were associated with metal ion homeostasis (GO:0055065), reactive oxygen species metabolic process (GO:0072593), cellular detoxification (GO:1990748), etc. (Supplementary Fig. 13c). These results suggest that improper Si deposition in the leaves of the mutant resulted in multiple responses similar to those caused by various stresses.

### Phylogenetic analysis of SIET4

We performed a phylogenetic analysis of Lsi2/SIET-like proteins in the plant kingdom. Homologs of Lsi2/SIET were found in the angiosperm including both monocots and dicots, *Polypodiophyta*, *Lycopodiophyta*, *Sphenopsida* and *Bryophyta*, but were not found in the gymnosperm, *Hepatopsida* and algae (Supplementary Fig. 14). Each plant species

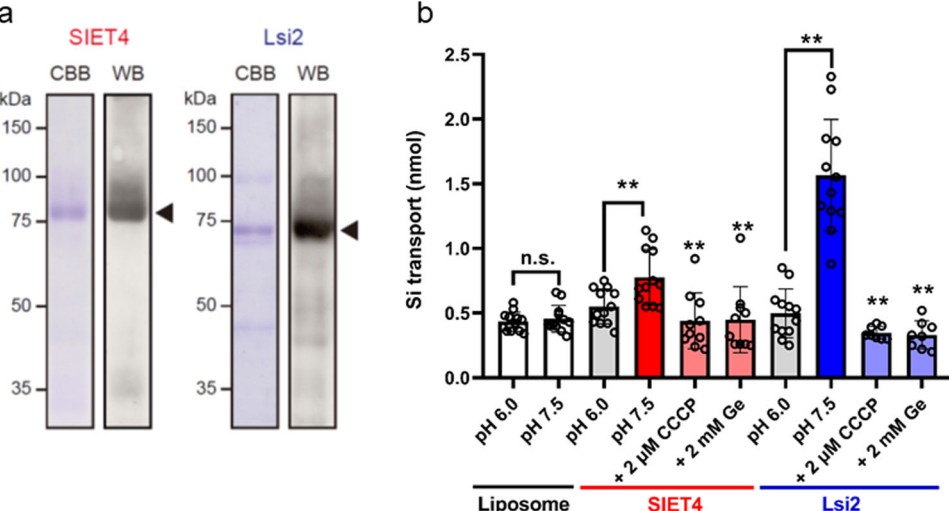

**Fig. 3 | Transport activity of SIET4 protein. a** Purification of SIET4 and Lsi2 proteins. The purified fraction was analyzed by SDS–PAGE and visualized by Coomassie Brilliant Blue staining (left) and by western blot with anti-6×His antibody (right). **b** SIET4- and Lsi2-mediated Si transport activity. The proteoliposomes containing SIET4 or Lsi2 were incubated in a buffer solution containing 1 mM Si at either pH 6.0 or pH 7.5, pH 7.5 plus 2 μM CCCP or pH7.5 plus 2 mM Ge, and assayed after 2 min. Si concentration was determined by ICP-MS. Data are mean ± s.e.m. of replicates of independent experiments ($n = 8$–12). *$P < 0.05$ and **$P < 0.01$ (two-tailed paired Student's $t$ test). The results obtained at pH 7.5 were compared with those at pH 7.5 + CCCP or pH 7.5 + Ge.

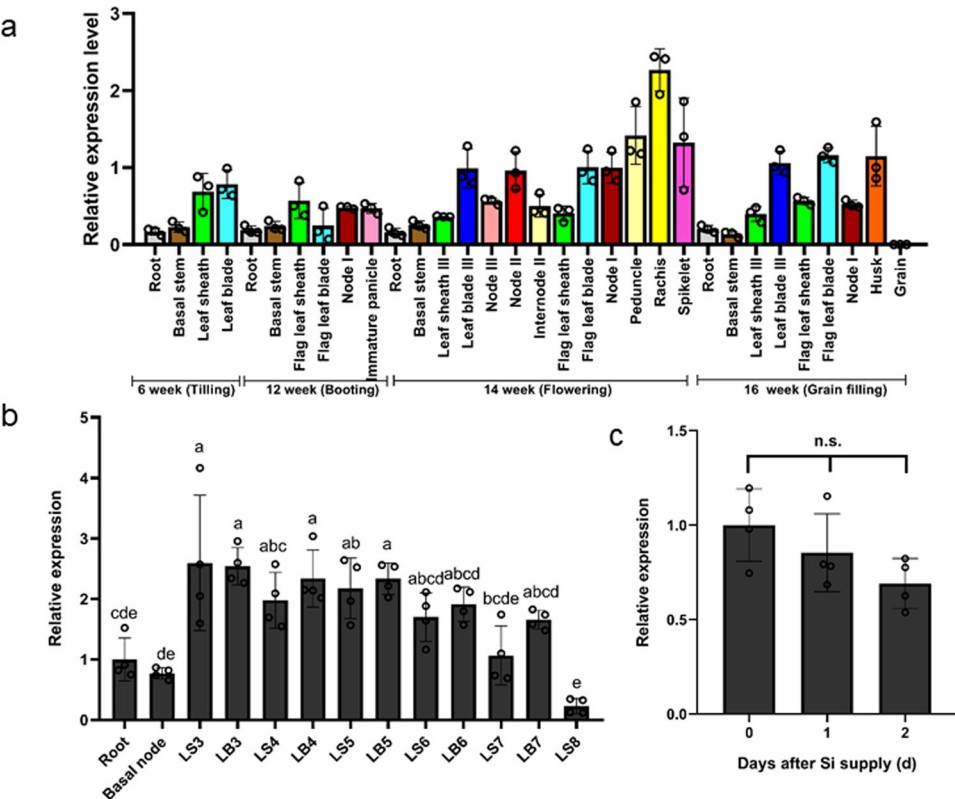

**Fig. 4 | Expression pattern of *SIET4*. a** Growth stage- and organ-dependent expression of *SIET4* in various organs at different growth stages. Different organs of rice grown in a paddy field were sampled at vegetative and reproductive stages and subjected to RNA extraction. **b** Leaf age-dependent expression pattern of *SIET4*. Seedlings (36-day-old) were cultured in a nutrient solution until appearance of leaf 8. Leaf-blade and leaf sheath from leaf 3 to leaf 7 were separately sampled for RNA extraction. LS, leaf sheath; LB, leaf blade; **c** Response of *SIET4* expression in the shoots to Si supply. Seedlings (22-day-old) were put in a nutrient solution containing 0 or 1 mM Si. At day 1 and 2, the shoot part was sampled for RNA extraction. The expression level was determined by quantitative RT-PCR. *Histone H3*, *Actin* and *Ubiquitin* were used as internal standards. Expression relative to Node I at the flowering stage (**a**), to root (**b**) and to -Si (**c**) is shown. Data are means ± SD ($n = 3$ for (**a**), $n = 4$ for (**b**, **c**) biologically independent plants). Different letters indicate significant difference ($P < 0.05$, **b**, **c**).

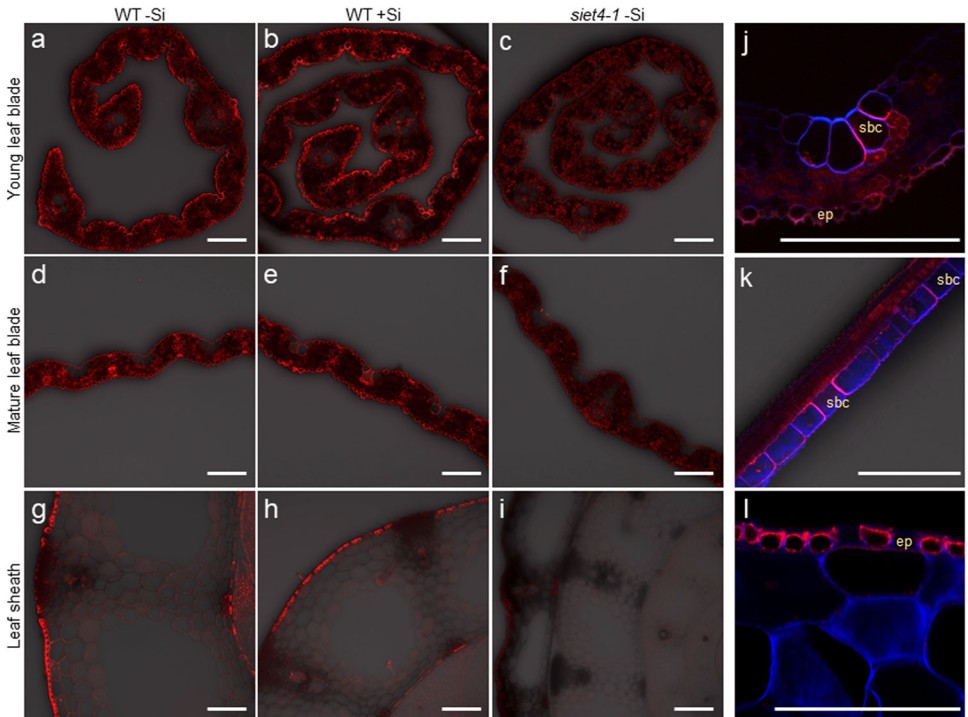

**Fig. 5 | Tissue and cellular localization of SIET4 in rice leaf. a–c** Localization of SIET4 in young leaf blade. **d–i**, Localization of SIET4 in leaf blade (**d–f**) and leaf sheath (**g–i**) of the mature leaf. Magnified image of cross-section (**j**) and longitudinal-section along by the bulliform cell layer (**k**) in mature leaf blade and leaf sheath (**l**). Seedlings of both wild-type rice (WT) (**a, b, d, e, g, h, j–l**) and *siet4-1* mutant (**c, f, i**) were put in a solution containing 0 (**a, c, d, f, g, i**) or 1 mM Si (**b, e, h, j–l**) for 1 day and the leaf blade and sheath were sampled for immunostaining with SIET4 antibody. The red color shows the signal from SIET4, and the blue color is from cell wall autofluorescence. ep, epidermis, sbc, silicified bulliform cells. Bar = 100 μm.

carries 1 to 8 copies of *Lsi2/SIET*-like genes, suggesting that *Lsi2/SIET* genes were acquired in primitive land plants and conserved in most land plants except gymnosperm and *Hepatopsida*.

Based on the similarity, the Lsi2/SIET type Si efflux transporter family could be roughly divided into two subgroups; Lsi2-like subgroup and SIET-like subgroup (Supplementary Fig. 14). The former subgroup includes Lsi2 and Lsi3 orthologs in monocots and all homologs in *Polypodiophyta*, *Lycopodiophyta*, *Sphenopsida* and *Bryophyta* (Supplementary Fig. 14), while the latter subgroup includes SIET3-5 orthologs in monocots and all homologs in dicots. Non-graminaceous monocot species, such as *Cyperaceae* and *Arecaceae* have intermediate homologs between the two subgroups. Interestingly, both subgroups are present only in graminaceous plants (Supplementary Fig. 14).

## Discussion

Rice is a typical Si-accumulating species and requires a large amount of Si for its healthy growth in soil[1,7,29]. To utilize Si, rice has developed an efficient system for Si accumulation, which starts from the uptake of Si by the roots from soil to the deposition in the different organs and tissues of above-ground parts[29,30]. Transporters involved in uptake, xylem loading/unloading, and distribution of Si have been identified in rice[8,9,23,25,31,32]. In the present study, through detailed functional analysis, we identified a key transporter, SIET4, which is required for the proper deposition of Si in the leaves of rice.

Si is considered to be the only mineral element that does not show excessive toxicity for the plants. This relies on its chemical properties and deposition pattern. Silicic acid {Si(OH)$_4$} is a non-charged molecule and will automatically polymerize to silica (SiO$_2$•nH$_2$O) when the concentration exceeds 2 mM[33]. However, Si as silicic acid in the xylem sap of rice may transiently exceed 10 mM (Supplementary Fig. 8b)[24,34,35]. Furthermore, after unloaded from the xylem, its concentration will become much higher in the leaf cells with water loss

through transpiration. Therefore, silicic acid is exported to the apoplast (metabolism inactive space) of specified cells of the leaves before the polymerization. We found that SIET4 functions as an efflux transporter of Si and is required for proper Si deposition in specific tissues and cells by exporting Si from leaf cells to the apoplastic space for final deposition in rice. This is supported by several lines of evidence: SIET4 is polarly localized at the epidermal cells and neighboring cells of bulliform cells of leaves (Fig. 5); knockout of this gene led to the death of rice in the presence of Si either in soil or nutrient solution with Si (Fig. 1a, b, d and Supplementary Fig. 2a, b), but did not affect the growth in the absence of Si (Fig. 1c and Supplementary Fig. 2a, b); and, knockout of *SIET4* resulted in Si deposition in the mesophyll cells in contrast to Si deposition on the epidermal cells of the WT (Fig. 2c, d).

Since knockout of *SIET4* failed to export Si to the apoplast of the epidermal cells, resulting in the accumulation of Si in mesophyll cells (Fig. 2c). This abnormal accumulation may cause various metabolism disturbances, finally resulting in plant death (Fig. 1 and Supplementary Fig. 4). This is supported by transcriptome analysis (Supplementary Fig. 13); Si supplied for 24 h induced the expression of more than one thousand genes in the leaves of the *siet4* mutant compared with the WT (Supplementary Fig. 13). These genes were related to those induced by various biotic and abiotic stresses, which are not usually induced in healthy plants. Abnormal expression of these genes may lead to unbalanced growth of rice, and finally to its death.

The importance of proper deposition of Si in apoplastic space in leaves is also well demonstrated by the ectopic expression of *Lsi1* from wheat and rice in Arabidopsis[15]. Arabidopsis is a non-Si-accumulating species because different from most other angiosperms, it exceptionally lacks Si influx transporter Lsi1[29,30]. When Lsi1 from rice or wheat was over-expressed in this species under the control of cauliflower mosaic virus 35S-promoter, Si uptake was significantly increased, but necrotic lesions on the leaves and reduced growth were systematically observed when the plant was grown in the solution with Si added

although growth was not reduced in the solution without Si added[15]. Si in the overexpressed lines was deposited at the base of trichomes, at the periphery of necrotic zones and around the stomata probably due to the universal expression of this gene under the control of the 35S-promoter. The improper Si deposition in the leaves caused growth inhibition[15].

Phylogenetic analysis showed that Lsi2/SIET transporter family was found in most land plants (Supplementary Fig. 14), but not in the gymnosperm, *Hepatopsida* and algae. This kind of transporter seems to be required for land plants to adapt to soils rich in Si, although their exact role in most plants remains to be examined. Since only graminaceous plants showing high Si accumulation possess genes belonging to both Lsi2-like and SIET-like subgroups (Supplementary Fig. 14), functional differentiation of the Lsi2/SIET transporter family may have occurred in the graminaceous plants; Lsi2/Lsi3 for Si uptake/long-distance-transport and SIETs for Si accumulation/deposition. Both are required for high and proper Si accumulation in the shoots of graminaceous plants.

In conclusion, our results show that SIET4 is required for the healthy growth of rice in soil. Its function is to export Si from leaf cells to the apoplastic space of epidermal cells to utilize Si. Whether this is a common strategy to cope with the abundant Si in soil in other land plants remains to be investigated.

## Methods

### Plant materials and growth conditions
The wild-type rice (WT, cv. Nipponbare) and two independent knockout lines of *SIET4* (*siet4-1* and *siet4-2*, T3/T4 generations) generated as described below, were used in this study. Seed germination and preparation of seedlings were as described previously[36]. The plants were grown in a controlled greenhouse at 25–30 °C, under natural light. All experiments were performed with at least three biological replicates.

### Generation of knockout lines of *SIET4* by CRISPR/Cas9
We generated knockout lines of *SIET4* by the CRISPR/Cas9 technique. Twenty bases upstream of the PAM motif were selected as candidate target sequences (Supplementary Fig. 1). The primers for two target sequences in the ORF region of SIET4 are listed in Supplementary Table 1. The plant expression vector of Cas9 (pU6gRNA) and single guide RNA expression vector (pZDgRNA_Cas9ver.2_HPT) were used as described before[37]. The derived constructs were transformed into rice calluses according to Hiei et al.[38].

To genotype the resultant mutants, we extracted genomic DNA from leaves of transgenic rice plants, followed by PCR amplification using primer pairs flanking the designed target sites as listed in Supplementary Table 1. The PCR products were sequenced directly using internal specific primers listed in Supplementary Table S1. Two independent homologous knockout lines without the Cas9 gene, *siet4-1* and *siet4-2* were selected for further analysis.

### Phenotypic analysis of *siet4* mutants in soil and hydroponic solution
For soil culture, seedlings (16-day-old) of WT and two mutants were grown in a half-strength Kimura B solution containing 1 mM Si as silicic acid prepared as described previously[39]. After another 24 days, the seedlings were transplanted to a 1/5000a Wagner pot containing 3.5 kg soil collected from the experimental field at Okayama University. The soil properties were as described previously[40] and Si concentration in the soil solution was between 0.6–0.8 mM during the growth period. Plants were watered daily and grown until maturation. At harvest, the shoot and panicles were separately sampled, and their dry weight was recorded after drying in an oven at 70 °C or 40 °C, respectively.

For hydroponic culture, the seedlings (16-day-old) were grown in a half-strength Kimura B solution containing 0 or 1 mM Si as silicic acid prepared as described previously[39]. Distilled water was used for

preparing the nutrient solution which contained less than 20 μg Si/L. The solution was changed once every two days. The biomass was monitored at different time points until heading. At the maturation stage, the roots, shoots, and panicles were separately sampled, and their dry weight was recorded as described above. During the growth period, the leaves were observed and photographed.

For a dose-response experiment, seedlings (27-day-old) of WT and *siet4* mutants were grown in a pot filled with 4.5 kg of river sand. Different rates (0, 10, 20, 50 and 100 g per pot) of Water Silica (Fuji Silysia Chemical Ltd.) were applied. The plants were grown in a closed greenhouse and supplied with Kimura B nutrient solution. Solution from the pot was collected several times during the growth period and subjected to Si determination as described below. After 50 days, the shoot parts were harvested and photographed. The shoot fresh weight was immediately recorded.

To determine the Si accumulation in the shoot of *siet4* mutants, we grew 1-month-old seedlings of WT and two mutants in pot soil as described above for 4 months until maturation or a nutrient solution containing 1 mM Si for 3 months in a greenhouse. Si concentration in the whole shoot was determined as described below.

SPAD values (total chlorophyll content) were determined on the fully expanded youngest leaves of WT and *siet4* mutants (20-day-old) treated with or without 1 mM Si for 5 days by using a portable chlorophyll meter (SPAD-502; Minolta Sensing).

For the determination of mineral concentration in *siet4* mutants and WT, seedlings (31-day-old) were put in a nutrient solution containing 5 μM germanium as $GeO_2$. After 24 h, the roots were washed with ice-cold 5 mM $CaCl_2$ solution three times and separated from the shoot parts. After dried at 70 °C, the samples of both roots and shoots were digested with concentrated $HNO_3$ (61%) at a temperature of up to 135 °C. Element concentrations were examined by inductively coupled plasma mass spectrometry (ICP-MS; 7700X; Agilent Technologies).

### Transport assay
The transport activity of SIET4 or Lsi2 as a positive control was assayed using proteoliposomes. The full length of SIET4 or Lsi2 was first cloned into β-pET-28a(+)-β, an expression vector for eukaryotic membrane proteins in *E. coli*, using the In-Fusion cloning kit (Takara). The procedures for the expression and purification followed the protocol as described previously[28]. For the transport assay, an aliquot (containing 30 μg) of purified SIET4 or Lsi2 was mixed with liposomes (500 μg) and frozen at −80 °C for at least 10 min. The mixture was diluted with reconstitution buffer containing 20 mM MES-KOH (pH 6.0), 0.1 M potassium acetate, and 5 mM magnesium acetate. Reconstituted proteoliposomes were pelleted by centrifugation at 200,000 g for 1 h at 4 °C and then suspended in reconstitution buffer. The 10 mg/mL liposomes containing 40% phosphatidylcholine, 30% phosphatidylethanolamine, 10% phosphatidylserine, and 20% cholesterol as weight ratio were prepared in the buffer containing 20 mM MES-KOH (pH 6.0) and 1 mM DTT as described previously[28]. For Si transport, reaction mixtures (130 μl) containing 0.5 μg of protein incorporated into proteoliposomes, 20 mM MES-KOH (pH 6.0) or 20 mM MOPS-KOH (pH 7.5), 0.1 M potassium acetate, 5 mM magnesium acetate, 10 mM KCl, and 1 mM Si as a silicic acid were incubated at 27 °C in the presence or absence of 2 μM CCCP or 2 mM Ge. According to our preliminary results, the transport activity was the highest at 2 min, therefore the transport assay was terminated after incubated for 2 min by separating the proteoliposomes from the external mixture using centrifuge columns containing Sephadex G-50 (fine). The Si incorporated into the liposomes was determined by inductively coupled plasma-mass spectrometer (ICP-MS) as described below.

A time-course experiment for SIET4-mediated transport of Si was also performed as described above. Briefly, the proteoliposomes with or without SIET4 at pH 6.0 inside were incubated in a buffer solution containing 1 mM Si at either pH 6.0 or pH 7.5. At indicated time points,

the proteoliposomes were sampled and subjected to determination of Si by ICP-MS. For the dose-dependent experiment of SIET4-mediated Si transport, the proteoliposomes with SIET4 at pH 6.0 inside were incubated in a buffer solution containing different Si concentrations at either pH 6.0 or pH 7.5. After incubation for 1.5 min, the proteoliposomes were sampled and subjected to determination of Si by ICP-MS as described below.

### Expression analysis of *SIET4*

To investigate the organ-dependent expression pattern of *SIET4* during the whole growth stage, we used cDNA of different organs as indicated in Fig. 4a, which was prepared in a previous study[41]. The leaf age-dependent expression of *SIET4* in the leaf blade, leaf sheath, basal node and roots harvested from 36-day-old seedlings grown in a nutrient solution without Si. The effect of Si was examined in seedlings kept in half-strength Kimura B solution with or without 1 mM Si for 1 or 2 days from day 22. The total RNA of leaves was extracted by using an RNeasy plant mini kit (QIAGEN) followed by cDNA synthesis according to the manufacturer of ReverTra Ace (TOYOBO). Specific cDNAs were amplified by Sso Fast EvaGreen Supermix (Bio-Rad) and quantitative real-time PCR was performed on CFX384 (Bio-Rad). *HistoneH3*, *Actin*, or *Ubiquitin* were used as internal controls. The primers used are listed in Supplementary Table 1. The relative expression was normalized based on internal control genes by ΔΔCt method using the CFX Manager software (Bio-Rad).

### Immunostaining of SIET4

To observe the tissue-specific localization and subcellular localization of SIET4, we performed immunostaining with an antibody against SIET4. The synthetic peptide C-RSNSVRSTSANENLRSR (positions 259–275 of SIET4) was used to immunize rabbits to obtain antibodies against SIET4. The obtained antiserum was purified through a peptide affinity column before use. Cross-sections of the leaf blade and leaf sheath of fully expanded leaf and immature leaf blade were prepared from one-month-old seedlings of WT and the knockout line with or without Si supplied for 1 day and subjected to immunostaining as described previously[42]. For observation of subcellular localization, an antibody for ER marker (HDEL antibody (2E7), Santa Cruz Biotechnology) and 4',6-diamidino-2-phenylindole (DAPI) for nucleic acid staining were used. The signal of fluorescence was observed by confocal laser scanning microscopy (TCS SP8x, Leica Microsystems).

### Uptake experiment of Si and xylem sap collection

To compare Si uptake between WT and the mutants, we conducted a short-term uptake experiment. Seedlings (26-day-old) of WT and two knockout mutants were put in a half-strength Kimura B solution containing 0.5 mM silicic acid. At 2, 4, 6, and 8 h, a part of the uptake solution was collected for determination of the Si concentration remaining in the uptake solution as described previously[23,39]. Water loss was also recorded at each sampling time point. At the end of the experiment, the roots and shoots were harvested separately and their fresh weights were recorded.

For collection of the xylem sap, seedlings (30-day-old) of WT and the two mutants grown in a nutrient solution free of Si were transferred to a solution containing 1 mM Si. After 2 h, the shoots were decapitated at 2 cm above the roots and the xylem sap was collected with a micropipette for 10 min. The sap was immediately diluted with distilled water and placed at room temperature overnight before Si determination.

### Determination of Si concentration

For determination of Si in plant tissues, the solution was digested in a microwave oven (Microwave Digestion System START D, Millstone Co., Ltd.) using a mixture of 4 ml of $HNO_3$ (62%), 4 ml of $H_2O_2$ (30%), and 1 ml of HF (46%)[43]. The digested solution was diluted to 50 ml with 4% (w/v) boric acid. The Si concentration in the digestion solution, uptake solution, xylem sap, and sand solution was determined by the colorimetric molybdenum blue method[44]. The Si concentration in the purified liposomes was determined by ICP-MS (7700X; Agilent Technologies) using the $H_2$-mode after dilution with 1 N $HNO_3$.

### Detection of Si deposition in leaf tissues by LA-ICP-MS

Si deposition in leaf tissues was detected by laser ablation-inductively coupled plasma-mass spectrometry (LA-ICP-MS). WT and *siet4* mutants were pretreated with or without Si for 6 d and the fully expanded leaf blades were sampled for analysis. Procedures for sample preparation and mineral element determination were the same as described previously[45]. At least two biological replicates of each sample were analyzed, which showed similar results.

### Si mapping and bulk quantification with SEM-EDX

To compare Si deposition on the leaf surface between different lines, we employed scanning electron microscopy-energy dispersive X-ray spectrometry (SEM-EDX). After the seedlings (36-day-old) were put in a solution containing 1 mM Si for 4, 8 or 24 h, the youngest expanded leaf blade was sampled with three biological replicates and the square area of its central part was used for observation using Miniscope TM3000 with SwiftED3000 (Hitach-hitec, Tokyo, Japan) in the surface observation mode (5 kV acceleration voltage). Si/C atomic ratio of the whole observation areas was measured.

### Transcriptome analysis

Seedlings (21-day-old) cultured in solution free of Si were treated with or without 1 mM Si for 24 h. Total RNA was extracted from the fully expanded leaf blade of both WT and *siet4* mutant using an RNeasy Plant Mini Kit (Qiagen). RNA-seq was performed using a DNBSEQ-G400FAST (MGI, Kobe, Japan) for paired-end sequencing. A total of 30 million to 40 million stranded paired-end (2×150 bp) sequences were obtained for each sample. Three biological replicates were made for each line and treatment. Sequences were mapped on IRGSP-1.0 rice reference genome (https://rapdb.dna.affrc.go.jp) and the FPKM (fragments per kilobase of exon per million mapped reads) values were compared using TopHat and Cufflinks on Galaxy/NAAC server (https://galaxy.dna.affrc.go.jp). Genes with a significant difference (>2-fold, <0.05 False Discovery Rate) in expression between the *siet4* mutant and WT with/without Si were extracted. The genes up- and down-regulated in the mutant were further used for gene ontology (GO) enrichment analysis, which was performed using PANTHER17.0 (http://pantherdb.org).

### Phylogenetic analysis

Amino acid sequences of Lsi2/SIET homologs in 55 plant species with available homolog sequences, representing 17 species/cultivars, were acquired from the database by BLAST search (https://blast.ncbi.nlm.nih.gov/). Sequence alignment by ClustalW and phylogenetic analysis by Maximum Likelihood method with 1000 bootstraps were conducted using MEGA X[46].

### Statistical analysis of data

Statistical analyses were performed by Student's *t* test or Tukey's-test using the software BellCurve for Excel (Social Survey Research Information Co. Ltd, Tokyo, Japan).

### Reporting summary

Further information on research design is available in the Nature Portfolio Reporting Summary linked to this article.

## Data availability

The experiment data that support the findings of this study are available from the corresponding author upon request. The RNA-seq data

generated in this study have been deposited to DDBJ BioProject under accession number PRJDB16702. The source data for Figs. 1 to 4 and Supplementary Figs 2, 3, 5, 6, 8, 9, 11 are provided as a Source Data file. Source data are provided with this paper.

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

## Acknowledgements

This work was supported by the Japan Society for the Promotion of Science (JSPS) (KAKENHI grant no. 16H06296, 21H05034 to JFM,

and 20K05773 to NM-U). We thank Narinobu Juge, Akemi Morita, Midori Hikasa, Yoshiko Murasawa, and Sanae Rikiishi for their assistance in the experiments. We also thank Dr. Masaki Endo for providing pU6gRNA and pZDgRNA_Cas9ver.2_HPT for the generation of CRISPR/Cas9 lines.

## Author contributions

N.M.-.U., N.Y., S.H. and J.F.M. designed the experiments and analyzed the data. N.M.-U., N.Y., S.H., Y.Y., T.M. and J.F.M. performed the experiments. N.M.-U., N.Y., S.H. and J.F.M. wrote the article. All authors discussed the results and read the manuscript.

## Competing interests

The authors declare no competing interests.
