## [Peer Review File · Nature Communications]

A silicon transporter gene required for healthy growth of rice
on landReviewer #1 (Remarks to the Author):

In this work, the authors demonstrate that knockout of the SIET4 gene in rice results in plants that are highly sensitive to Si, to the point of lethality. Knockout lines show distorted Si deposition in leaves, with aberrant accumulation in mesophyll cells resulting in tremendous stress. Proteoliposomes containing SIET4 demonstrate some Si-transport activity. Overall, the authors make a rather convincing case that SIET4 is necessary for rice to survive under high-Si conditions; this being said, I have some concerns, particularly related to the wider claim that SIET4 is "essential" to land plant evolution, as well as aspects specific to Si transport.

General comments

- The title, and the overall tenor of the ms, appear overreaching beyond what is supported by the evidence, namely in the inference beyond rice. The authors do not make a convincing case to have discovered a gene "essential" for the survival of plants on land. As the authors note (Ln 48), the emergence of embryophytes on land likely occurs in the mid-Ordovician (~470 Ma). This was prior to the evolution of diatoms, the major silicifying oceanic species (~100 Ma; Benoitson et al., 2017 (<https://doi.org/10.1098/rstb.2016.0397>)), when the world's oceans were nearly saturated with silicic acid (c. 1.7 mM; Conley et al., 2017 (<https://doi.org/10.3389/fmars.2017.00397>); Berges et al., 2022 (<https://doi.org/10.1016/B978-0-12-822861-6.00023-6>)). Thus, it is not evident that plants had to "cope" (Ln 25) with silicon when migrating from sea to land in order to survive. Indeed, present day silicic acid concentrations in seawater may be limited (averaging <70 μ M; Berges et al., 2022), but they are not much higher on land (averaging c. 100 – 600 μ M; Epstein, 2001 ([https://doi.org/10.1016/S0928-3420\(01\)80005-7](https://doi.org/10.1016/S0928-3420(01)80005-7))). It would behoove the authors to make the distinction that, although Si is indeed the most abundant mineral element in soils (Ln 26, 50-1, 241-2, etc.), the vast majority is biologically inert silica.
- There is a general need for a thorough English language editing.

Re Results

- Ln 158ff – The data related to SIET4-mediated Si transport are somewhat puzzling. To my knowledge, this is the first such demonstration of a Si-transport assay for plant Si transporters in proteoliposomes. The authors cite Miyaji et al., 2015 in reference to this method; however, this article describes an ascorbate transporter. The results of the Si transport assay (Fig. 3) suggest some Si uptake in proteoliposomes; however, much more evidence is required to offer proof of its transport function. For one, it would be much more convincing that SIET4 was a bona-fide Si transporter if parallel results (i.e., positive controls) were presented for previously characterized plant Si transporters (e.g., the homolog Lsi2). Moreover, a time course and external Si concentration-dependent flux analysis would be highly desirable; even more so were competition data with Ge to be presented. Lastly, if SIET4 is indeed responsible for releasing Si into the leaf apoplast (e.g., Ln 40), and related to Lsi2 (Ln 99, 220ff), the authors should provide evidence of Si efflux (the data from Fig. 3 only suggest influx), and, ideally, evidence for Si/H⁺ antiport, as the authors themselves have previously demonstrated for Lsi2.

Ln 199ff: The authors should address the transcriptomic data along with the phenotypic data with both more context and more circumspection as they appear at odds with some statements made by the senior author in previous publications. In this work, the authors observed nearly 550 genes that were significantly regulated (up or down) by Si in WT plants, which contradicts some previous work as written in a review paper co-authored by Professor JF Ma: "Watanabe et al. (2004) first demonstrated that Si supplementation had essentially no effects on gene expression in rice, with only one of c. 9000 genes analyzed being significantly altered in its expression. Similarly, a proteomic analysis by Nwugo & Huerta (2011) showed that Si supplementation in rice resulted in statistically significant changes in the abundance of only four proteins, and another study with rice could not identify distinct metabolic pathways influenced by Si in control plants (Brunings et al., 2009)." In the same manner, the authors report a nearly 2-fold increase in shoot dry weight in WT plants as a results of Si treatment, while there is a large body of work that has shown no to minimal effect of Si in control plants in many plant species. While the results obtained in this work are not contested (but should be contextualized), contradictions obtained in other works and with other plants preclude, unless scientific evidence is obtained, from making a sweeping generalization that SIET4 is "essential for survival of plants on land." They support the case for rice only and the title and conclusion should reflect this limitation.

-

Re Discussion

- Ln 252ff – This paragraph is simply repeating the results and is thus redundant.

Reviewer #2 (Remarks to the Author):

Dear Authors,

Your article is well written and easy to read. I regret that it contains some inaccuracies and information is missing.

The title is very catchy but unfortunately I did not clearly find the results which allowed to reach this conclusion.

<please see comments in attached files>

In their review of the first version of this manuscript, reviewer #2 added some comments to the manuscript file. These comments were forwarded to the authors, who replied as included in this Peer Review File.

Reviewer #3 (Remarks to the Author):

Comments on manuscript entitled "A gene essential for survival of plants on land"

The manuscript entitled "A gene essential for survival of plants on land" by Mitani-Ueno et al., reported a new gene (SIET4) which is polarly localized at the distal side of epidermal cells and cells and constitutively expressed in leaves throughout the whole period of rice growth. The authors claimed that this gene encoding for a plasma membrane-localized transporter of Si is responsible for Si deposition on leaves of rice, but not involved in the uptake and translocation of Si. However, SIET has Si transport activity. Knockout of SIET4 led to death of rice in the presence of Si, but did not affect the growth in the solution with no exogenous Si added. Knockout of SIET4 induced abnormal Si deposition in the mesophyll cells, resulting in induction of hundreds of genes involved in various biotic/abiotic stress responses. Based on these observations and phylogenetic tree, the authors concluded that this gene is essential for survival of plants on land. In general, this manuscript dealing with identification of a new gene and analysis of its possible functions is very interesting, helping understand Si depositions on rice leaves and the effect of such deposition on the growth of rice. This manuscript is original and novel, in my opinion, it could thus be reconsidered for publication in Nature Communications after major and thorough revisions.

Major concerns:

1) In the solution with no exogenous Si added, the growth of rice plants lacking SIET4 (knockout of SIET4) was not affected. This treatment was improperly claimed by the authors as without Si (in the absence of Si). Actually, Si concentration could be as high as about 0.005-0.008 mg/L even in double-distilled water as reported by Woolley (Woolley, J. T. 1957. Plant Physiol. 32, 317-321). Woolley (1957) went to considerable lengths to exclude Si from the environment to determine whether it is an essential element for the tomato, *Lycopersicon esculentum*. The Si-deprived plants grew as well as those deliberately supplied with Si. The Si content of the shoots of the Si-deprived plants was 4.2 ppm, and that of the roots, 2.8 ppm. These values are on the order of levels of Cu generally considered adequate. So, no "Si-free" environment can be created due to contamination from distilled water,

chemicals used for solution culture and dust in the growth chamber (see Epstein, 1994, Epstein, 1999). We doubt if there is a critical or threshold value of Si above which the growth of rice plants lacking SIET4 (knockout of SIET4) is abnormal or close to death. So, the authors need to supplement a solution culture experiment with a series of Si level (e.g. 0, 0.001, 0.01, 0.05, 0.1, 0.5, 1.0 mM Si added) designed to determine the performance of WT, SIET4-1 and SIET4-2 rice plants and find out the critical value above which the rice mutants lacking SIET4 (knockout of SIET4) died or almost died. If so, the conclusion in this MS could not be supported.

2) In my opinion, the title of the manuscript is not appropriate, not supported by the results provided in this study. Is the gene (SIET4) essential for survival of all plants on land? The authors observed that rice plants lacking SIET4 grew abnormally in solutions with Si added and in soil, while no such trials were performed on other monocots or dicots. Phylogenetic analysis of SIET4 showed that only in graminaceous plants, both subgroups (Lsi2-like subgroup and SIET-like subgroup) are present (Extended Data Fig. 11), indicating that quite a lot of plants do not contain SIET-like genes but contain Lsi2-like genes which are not responsible for Si deposition but associated with Si transport from cells to xylem (Si efflux). Actually, whether the gene (SIET4) is essential for survival of rice on land could not be concluded in this manuscript (see the comment above).

3) In Figure 2, no data on Si concentration in plants grown in solution with no Si added was present. These data are very important, and thus are needed for reference. As we know, rice plants cannot grow as well (tillering and fertilizing) in solutions with no exogenous Si added as in solutions with 1.0 mM Si added because the amount of Si from distilled water, chemical agents and dust in solutions with no exogenous Si added cannot satisfy the requirements of Si for normal growth of rice, an Si-hyperaccumulator). However, the growth of phenotypes of either WT or mutants lacking SIET was relatively normal in solution with no Si added in this study. This is a phenomenon difficult to understand.

4) The authors only grew WT and SIET line in the rice field on the university campus, but not in other different soils with very low to high Si availability (as indicated by plant available Si). It can be expected that maybe in some soils with extremely low plant-available Si both WT and SIET plants can grow relatively well or normally with no plant death observed. If so, the major conclusion as claimed by the title of the manuscript could not be drawn (see also comment 1).

5) With the solution and field trial results, the authors observed that SIET lines both in solution with 1.0 mM Si added and in soil died. However, we are still unknown which factor(s) is the direct cause leading to the death of SIET lines both in solution with 1.0 mM Si added and in soil although knockout of SIET induced a lot of alterations in gene expressions (Quite some were up-regulated and others were down-regulated). It can be expected that knockout of LSi1, LSi2 or LSi3 or LSi6 as reported by the authors will result in similar changes in gene expressions. We are unclear if knockout of other 1 or 2 Si transporters except SIET will lead to the death of the mutants in soil or in solutions with Si added. On the other hand, in LSi1 (GR) mutant grown hydroponically with 1.5 mM Si added or in potted soil, the concentration of Si in shoots could be as high as 1.43-1.49% (very high) (see Ma et al., 2002. *Plant Physiol.* 130:2111-2117). The same might be applied to other mutants (e.g. LSi2). That means that Si could be taken up via other pathways (e.g. via transpiration streams, passive uptake). Is the Si transported via transpiration streams deposited on rice leaves through SIET transporter? We have no ideas at the moment.

6) How were plant roots pretreated before drying in oven? No details were given in the MM section. If roots were not pretreated in a traditional way, the contents of divalent and trivalent cations in the roots would be overestimated. These data are not acceptable. I am sure that the iron plaque on the surface of roots was not washed completely because the concentration of Fe (percent) in roots was several orders of magnitude higher than accepted (Iron is a micronutrient) (see Extended Data Fig. 7) .

7) Need to give a detailed description of soil physio-chemical properties including the concentration of plant-available Si in the soil (on the university farm) used for field trial.

Minor concerns:

1) The readability is overall good. However, there are many grammatical English errors. So, the authors need to invite a native speaker (professional the best) to go through the manuscript carefully. English needs polishing carefully.

We thank all reviewers for their positive and constructive comments!. We have revised our manuscript as detailed below.

REVIEWER COMMENTS

Reviewer #1 (Remarks to the Author):

In this work, the authors demonstrate that knockout of the SIET4 gene in rice results in plants that are highly sensitive to Si, to the point of lethality. Knockout lines show distorted Si deposition in leaves, with aberrant accumulation in mesophyll cells resulting in tremendous stress. Proteoliposomes containing SIET4 demonstrate some Si-transport activity. Overall, the authors make a rather convincing case that SIET4 is necessary for rice to survive under high-Si conditions; this being said, I have some concerns, particularly related to the wider claim that SIET4 is “essential” to land plant evolution, as well as aspects specific to Si transport.

General comments

- The title, and the overall tenor of the ms, appear overreaching beyond what is supported by the evidence, namely in the inference beyond rice. The authors do not make a convincing case to have discovered a gene “essential” for the survival of plants on land. As the authors note (Ln 48), the emergence of embryophytes on land likely occurs in the mid-Ordovician (~470 Ma). This was prior to the evolution of diatoms, the major silicifying oceanic species (~100 Ma; Benoitson et al., 2017 (<https://doi.org/10.1098/rstb.2016.0397>)), when the world’s oceans were nearly saturated with silicic acid (c. 1.7 mM; Conley et al., 2017 (<https://doi.org/10.3389/fmars.2017.00397>); Berges et al., 2022 (<https://doi.org/10.1016/B978-0-12-822861-6.00023-6>)). Thus, it is not evident that plants had to “cope” (Ln 25) with silicon when migrating from sea to land in order to survive. Indeed, present day silicic acid concentrations in seawater may be limited (averaging <70 μ M; Berges et al., 2022), but they are not much higher on land (averaging c. 100 – 600 μ M; Epstein, 2001 ([https://doi.org/10.1016/S0928-3420\(01\)80005-7](https://doi.org/10.1016/S0928-3420(01)80005-7))). It would behoove the authors to make the distinction that, although Si is indeed the most abundant mineral element in soils (Ln 26, 50-1, 241-2, etc.), the vast majority is biologically inert silica.

Response: Thank you for your comments! We agree with your comments and have revised the manuscript by focusing on rice. Accordingly, we have changed the title and reconsidered the text.

- There is a general need for a thorough English language editing.

Response: We have checked the language thoroughly.

Re Results

- Ln 158ff – The data related to SIET4-mediated Si transport are somewhat puzzling. To my knowledge, this is the first such demonstration of a Si-transport assay for plant Si transporters in proteoliposomes. The authors cite Miyaji et al., 2015 in reference to this method; however, this article describes an ascorbate transporter. The results of the Si transport assay (Fig. 3) suggest some Si uptake in proteoliposomes; however, much more evidence is required to offer proof of its transport function. For one, it would be much more convincing that SIET4 was a bona-fide Si transporter if parallel results (i.e., positive controls) were presented for previously characterized plant Si transporters (e.g., the homolog Lsi2). Moreover, a time course and external Si concentration-dependent flux analysis would be highly desirable; even more so were competition data with Ge to be presented. Lastly, if SIET4 is indeed responsible for releasing Si into the leaf apoplast (e.g., Ln 40), and related to Lsi2 (Ln 99, 220ff), the authors should provide evidence of Si efflux (the data from Fig. 3 only suggest influx), and, ideally, evidence for Si/H⁺ antiport, as the authors themselves have previously demonstrated for Lsi2.

Response: We have performed several additional experiments on SIET4 transport activity in response to your comments. Proteoliposome method is a well established one for determining transport activity of a transporter. We followed basic procedures reported previously in Miyaji et al., 2015, but we have added more information on Si transport activity in the revised manuscript.

Proteoliposome method allows us to determine both influx and efflux transport activity by changing pH inside and outside of the liposome. We performed several experiments and found that 1), similar to Lsi2, SIET4 showed an efflux transport activity for Si, 2), Ge was able to inhibit Si uptake by SIET4, 3), when the pH between inside and outside of liposome was same, SIET4 lost its transport activity, whereas when the pH outside the liposome is higher than that inside, the transport activity was observed, indicating that SIET4 is an antiporter with proton, 4) this is further confirmed by the experiment with CCCP, an ionophore inhibitor.

Since Proteoliposome method is an in vitro one, different from in vivo experiments, it is not suitable for time-course and dose-response experiments. Furthermore, Si was polymerized at high Si concentration, while it was too low to be detected at low Si concentration.

We have added all additional results in the revised manuscript.

Ln 199ff: The authors should address the transcriptomic data along with the phenotypic data with both more context and more circumspection as they appear at odds with some

statements made by the senior author in previous publications. In this work, the authors observed nearly 550 genes that were significantly regulated (up or down) by Si in WT plants, which contradicts some previous work as written in a review paper co-authored by Professor JF Ma: “Watanabe et al. (2004) first demonstrated that Si supplementation had essentially no effects on gene expression in rice, with only one of c. 9000 genes analyzed being significantly altered in its expression. Similarly, a proteomic analysis by Nwugo & Huerta (2011) showed that Si supplementation in rice resulted in statistically significant changes in the abundance of only four proteins, and another study with rice could not identify distinct metabolic pathways influenced by Si in control plants (Brunings et al., 2009).” In the same manner, the authors report a nearly 2-fold increase in shoot dry weight in WT plants as a results of Si treatment, while there is a large body of work that has shown no to minimal effect of Si in control plants in many plant species. While the results obtained in this work are not contested (but should be contextualized), contradictions obtained in other works and with other plants preclude, unless scientific evidence is obtained, from making a sweeping generalization that *SIET4* is “essential for survival of plants on land.” They support the case for rice only and the title and conclusion should reflect this limitation.

Response: Beneficial effects of Si is characterized by mitigating various stresses in plants. Therefore, gene expression changes differ with growth conditions, plant age, Si treatment period, etc. In the present study, in order to investigate initial response of *siet4* mutant to Si, we exposed the plants to Si only for 24 h, rather than long term as reported in other studies. This may be the main reason why the genes differentially expressed are different between different studies. The other reason is the sensitivity between different methods. We used RNA-seq in this study, which has much higher sensitivity compared with microarray reported previously (e.g. Watanabe et al., 2004).

Since the aim of this experiment is to understand why knockout of *SIET4* results in growth reduction in the presence of Si, we only focused on genes induced in the mutant, but did not discuss about the genes involved in beneficial effects of Si. However, we have made some discussion on this aspect in the revised manuscript.

-

Re Discussion

- Ln 252ff – This paragraph is simply repeating the results and is thus redundant.

Response: We have deleted this paragraph.

Reviewer #2 (Remarks to the Author):

Dear Authors,

Your article is well written and easy to read. I regret that it contains some inaccuracies and information is missing.

The title is very catchy but unfortunately I did not clearly find the results which allowed to reach this conclusion.

Response: Thank you for your constructive comments! We have changed the title and addressed your comments in the pdf file you made comments.

Reviewer #3 (Remarks to the Author):

Comments on manuscript entitled "A gene essential for survival of plants on land"

The manuscript entitled "A gene essential for survival of plants on land" by Mitani-Ueno et al., reported a new gene (SIET4) which is polarly localized at the distal side of epidermal cells and cells and constitutively expressed in leaves throughout the whole period of rice growth. The authors claimed that this gene encoding for a plasma membrane-localized transporter of Si is responsible for Si deposition on leaves of rice, but not involved in the uptake and translocation of Si. However, SIET has Si transport activity. Knockout of SIET4 led to death of rice in the presence of Si, but did not affect the growth in the solution with no exogenous Si added. Knockout of SIET4 induced abnormal Si deposition in the mesophyll cells, resulting in induction of hundreds of genes involved in various biotic/abiotic stress responses. Based on these observations and phylogenetic tree, the authors concluded that this gene is essential for survival of plants on land. In general, this manuscript dealing with identification of a new gene and analysis of its possible functions is very interesting, helping understand Si depositions on rice leaves and the effect of such deposition on the growth of rice. This manuscript is original and novel, in my opinion, it could thus be reconsidered for publication in Nature Communications after major and thorough revisions.

Major concerns:

1) In the solution with no exogenous Si added, the growth of rice plants lacking SIET4 (knockout of SIET4) was not affected. This treatment was improperly claimed by the authors as without Si (in the absence of Si). Actually, Si concentration could be as high as about 0.005-0.008 mg/L even in double-distilled water as reported by Woolley (Woolley, J. T. 1957. Plant Physiol. 32, 317-321). Woolley (1957) went to considerable lengths to exclude Si from the environment to determine whether it is an essential element for the tomato, *Lycopersicon*

esculentum. The Si-deprived plants grew as well as those deliberately supplied with Si. The Si content of the shoots of the Si-deprived plants was 4.2 ppm, and that of the roots, 2.8 ppm. These values are on the order of levels of Cu generally considered adequate. So, no “Si-free” environment can be created due to contamination from distilled water, chemicals used for solution culture and dust in the growth chamber (see Epstein, 1994, Epstein, 1999). We doubt if there is a critical or threshold value of Si above which the growth of rice plants lacking SIET4 (knockout of SIET4) is abnormal or close to death. So, the authors need to supplement a solution culture experiment with a series of Si level (e.g. 0, 0.001, 0.01, 0.05, 0.1, 0.5, 1.0 mM Si added) designed to determine the performance of WT, SIET4-1 and SIET4-2 rice plants and find out the critical value above which the rice mutants lacking SIET4 (knockout of SIET4) died or almost died. If so, the conclusion in this MS could not be supported.

Response: Thank you for your detailed comments! We agree that it is impossible to remove Si completely from the nutrient solution because Si is very rich in the environment. We used distilled water for preparing –Si nutrient solution, which only contained about 20 ug/L. Since this work focuses on high Si deposition in rice, which is a Si-accumulating plant, we think that a tiny amount of Si contamination will not affect our conclusion.

In response to your comments, we performed a dose-response experiment by using Si added ranging from 0 to 2 mM. As shown in Supplementary Fig. 6, 7, addition of Si concentration at 0.1 mM for one month did not affect the growth of *siet4* mutants, whereas addition of higher Si concentrations resulted in decreased growth and appearance of toxic symptoms in *siet4* mutants. Since Si in soil solution is usually higher than 0.1 mM, this means that SIET4 is required for normal growth in soil.

2) In my opinion, the title of the manuscript is not appropriate, not supported by the results provided in this study. Is the gene (SIET4) essential for survival of all plants on land? The authors observed that rice plants lacking SIET4 grew abnormally in solutions with Si added and in soil, while no such trials were performed on other monocots or dicots. Phylogenetic analysis of SIET4 showed that only in graminaceous plants, both subgroups (Lsi2-like subgroup and SIET-like subgroup) are present (Extended Data Fig. 11), indicating that quite a lot of plants do not contain SIET-like genes but contain Lsi2-like genes which are not responsible for Si deposition but associated with Si transport from cells to the xylem (Si efflux). Actually, whether the gene (SIET4) is essential for survival of rice on land could not be concluded in this manuscript (see the comment above).

Response: We agree with your opinion and have changed the title and other parts by focusing on rice. We also made some discussion based on the phylogenetic analysis.

3) In Figure 2, no data on Si concentration in plants grown in solution with no Si added was present. These data are very important, and thus are needed for reference. As we know, rice plants cannot grow as well (tillering and fertilizing) in solutions with no exogenous Si added as in solutions with 1.0 mM Si added because the amount of Si from distilled water, chemical agents and dust in solutions with no exogenous Si added cannot satisfy the requirements of Si for normal growth of rice, an Si-hyperaccumulator). However, the growth of phenotypes of either WT or mutants lacking SIET was relatively normal in solution with no Si added in this study. This is a phenomenon difficult to understand.

Response: As mentioned above, beneficial effects of Si could be observed only under stress condition. Since we grew the plants in an environment-controlled room with much less stress, the plants could grow well without Si addition.

We also tried to determine the Si concentration in the shoot of WT and *siet4* mutants grown in -Si solution, however, the concentration was too low to be detected by the colorimetric Molybdenum blue methods. We have mentioned this in the result section.

4) The authors only grew WT and SIET line in the rice field on the university campus, but not in other different soils with very low to high Si availability (as indicated by plant available Si). It can be expected that maybe in some soils with extremely low plant-available Si both WT and SIET plants can grow relatively well or normally with no plant death observed. If so, the major conclusion as claimed by the title of the manuscript could not be drawn (see also comment 1).

Response: Yes, it is well-known that Si concentrations in soil solution differ greatly with soil type. But in most soils, the Si concentration is higher than 0.1 mM. According to our dose-response experiment result shown in Supplementary Fig. 6, Si at a concentration higher than 0.1 mM negatively affected the growth of *siet4* mutants. Furthermore, Si fertilizers are routinely applied to paddy field to sustain high rice yield, further supporting that SIET4 is required for rice growth in most soils.

5) With the solution and field trial results, the authors observed that SIET lines both in solution with 1.0 mM Si added and in soil died. However, we are still unknown which factor(s) is the direct cause leading to the death of SIET lines both in solution with 1.0 mM Si added and in soil although knockout of SIET induced a lot of alterations in gene expressions (Quite some were up-regulated and others were down-regulated). It can be expected that knockout of LSi1, LSi2 or LSi3 or LSi6 as reported by the authors will result in similar changes in gene expressions. We are unclear if knockout of other 1 or 2 Si transporters except SIET will lead

to the death of the mutants in soil or in solutions with Si added. On the other hand, in LSi1 (GR) mutant grown hydroponically with 1.5 mM Si added or in potted soil, the concentration of Si in shoots could be as high as 1.43-1.49% (very high) (see Ma et al., 2002. Plant Physiol. 130:2111-2117). The same might be applied to other mutants (e.g. LSi2). That means that Si could be taken up via other pathways (e.g. via transpiration streams, passive uptake). Is the Si transported via transpiration streams deposited on rice leaves through SIET transporter? We have no ideas at the moment.

Response: Part of Si may be taken up by other pathways, but the contribution will be very small because knockout of either Lsi1 or Lsi2 significantly reduced Si uptake as we reported previously. Different from Lsi1, Lsi2 and Lsi6, which are mainly involved in Si uptake and preferential distribution, SIET4 is responsible for the deposition of Si in the leaves. Different from other Si mutants, knockout of *SIET4* led to the death. As we discussed, this is probably caused by improper deposition of Si in the leaves. Failure of Si export to the apoplast in the mutants results in accumulation of Si in the cytosol, which acts as a stress signal. This is supported by our transcriptome analysis; hundreds of genes related to stress are up-regulated in the mutants in the presence of Si.

6) How were plant roots pretreated before drying in oven? No details were given in the MM section. If roots were not pretreated in a traditional way, the contents of divalent and trivalent cations in the roots would be overestimated. These data are not acceptable. I am sure that the iron plaque on the surface of roots was not washed completely because the concentration of Fe (percent) in roots was several orders of magnitude higher than accepted (Iron is a micronutrient) (see Extended Data Fig. 7) .

Response: We have described in M&M that roots were washed with the ice-cold 5 mM CaCl₂ for 3 times as established in many studies. After blotting with a paper towel, the samples were dried in an oven at 70 degree for more than 2 days. Since we used ferrous iron in the nutrient solution, a major Fe form in paddy field, the Fe concentration in the roots was very high, but we did not remove the iron plaque.

7) Need to give a detailed description of soil physio-chemical properties including the concentration of plant-available Si in the soil (on the university farm) used for field trial.

Response: We have cited a paper for physio-chemical properties of soil used. The Si concentration in soil solution was around 0.6-0.8 mM during the growth period.

Minor concerns:

1) The readability is overall good. However, there are many grammatical English errors. So,

the authors need to invite a native speaker (professional the best) to go through the manuscript carefully. English needs polishing carefully.

Response: We have polished the English as indicated.

Reviewer #1 (Remarks to the Author):

Re response letter:

- Re proteoliposome assay:

o Contrary to the authors' claims, there is no evidence provided for SIET4-mediated Si efflux in their proteoliposome assay (Fig. 3). Thus, the claim that SIET4 mediates Si efflux remains unsubstantiated.

o What is the evidence for internal pH that would support the claim of a pH gradient (or lack thereof) across the proteoliposome? As far as this reviewer can tell, these claims are also unsupported.

o The authors argue that in-vitro assays like the one described here are not suited to dose- and time-course analyses, but this reviewer would argue otherwise. If this were indeed the case, how can the authors justify their choice of assay conditions (i.e., 1 mM Si over 2 min)? Moreover, in another in-vitro assay typical of Si-transporter studies, time- and dose-dependent curves are routinely conducted.

Re main text:

- Ln 35-36: In line with the comments made by another reviewer, "the absence of Si" is misleading as there are no feasible experimental conditions wherein all Si is removed. A term such as "Si supplementation" (or lack thereof) should be considered (see also Ln 112, 120, etc.)

- Ln 50: I disagree that "all plants contain significant amount of Si in the above-ground part". As the authors themselves have noted over the years, plants can be categorized as "passive or rejective" in terms of their Si accumulation – surely these cases would not be considered "significant".

- Light language editing still required throughout the ms.

- Ln 297: This argument requires clarification. If all dicots contain SIET4 (Ln 257), why the "improper" Si deposition in transgenic Arabidopsis?

- Fig 3:

o Caption should specify what the statistical test results relate to for CCCP and Ge treatments.

o Again, the data only demonstrate influx, not efflux, contradicting the response letter, and thus failing to support the claim that SIET4 mediates cellular Si efflux.

o This data is further complicated by the fact that the authors described Lsi2 as unidirectional (efflux only) in a previous report (albeit in another heterologous system, i.e., *Xenopus* oocytes; Ma et al., 2007), but demonstrate Lsi2-mediated influx here. This discrepancy should be addressed.

Reviewer #3 (Remarks to the Author):

The revised MS has been improved significantly. However, There are still some points to be addressed.

1) In the Supplementary Figure 6, the shoot biomass of mutant SIET 4-1 grown hydroponically with 0.1, 0.3 or even 1.0 mM Si added was insignificantly different from that of CK, suggesting that the mutant can grow well in solutions with 0.1-1.0 mM Si added like WT. Since the Si concentration in soil solutions normally ranges from 0.1-0.6 mM, we can guess that at a very high concentration of Si, the mutants (SIET 4-1 and SIET 4-2) can survive without significant loss of biomass (especially the shoot biomass) during 21-day growth period. If the authors did similar pot trials with different types of soils containing 0.1-1.0 mM Si in soil solutions with mutants and WT, the mutants would hopefully grow as well as WT. If so, the conclusions as claimed in the title, abstract and conclusion sections cannot be fully supported, i.e. this gene is not essential for survival of rice in all soils. I still believe that there is a threshold or critical value below which the SIET mutant can grow well in soils as WT. We need more evidence using soil culture experiment. From the results of Supplementary Figure 6, it is clear that the two mutants, especially SIET 4-1, can grow well at a very high concentration of Si (0.3-1.0 mM). I suggest that the authors do a supplementary experiment with soils differing in soil solution Si level (e.g. 0.1, 0.2, 0.5, 1.0 and 2.0 mM) to find out the critical value. This information is critical because it can modify the major conclusion of this MS or make the conclusion more correct or rational.

2) We are still unknown that what is the direct reason to cause plant death for the mutants. I suggest that the authors need to explain in more details in discussion.

3) The English needs polishing by a native speaker because there are still some English errors.

We thank all reviewers for further comments, which are important for improving our paper. We have further revised the paper by performing several experiments as detailed below.

Re response letter:

- Re proteoliposome assay:

o Contrary to the authors' claims, there is no evidence provided for SIET4-mediated Si efflux in their proteoliposome assay (Fig. 3). Thus, the claim that SIET4 mediates Si efflux remains unsubstantiated.

Response: The results shown in Fig. 3 is efflux activity, rather than influx activity. Proteoliposome assay is a well-established method for determination of transport activity. Since the transporter protein is bi-directionally reconstituted into liposomes in this assay system, this allows us to determine either efflux or influx activity depending on pH inside and outside the liposome. In this study, we found that the Si transport activity of SIET4 was only detected when the pH inside the liposome is 6.0 (similar to apoplasmic pH), while the pH outside the liposome is 7.5 (similar to cytosol pH). This means that Si by SIET4 is transported from cytosol to the apoplast, indicating that SIET4 is an efflux transporter for Si. We have added more explanation on this method in the revised manuscript.

o What is the evidence for internal pH that would support the claim of a pH gradient (or lack thereof) across the proteoliposome? As far as this reviewer can tell, these claims are also unsupported.

Response: As we described in the method part, we first prepared proteoliposomes in a buffer at pH 6.0, followed by suspending in a buffer at same pH (6.0). Therefore, the pH inside the liposome is kept at pH 6.0. By changing pH in the assay solution, we can generate the pH gradient.

o The authors argue that in-vitro assays like the one described here are not suited to dose- and time-course analyses, but this reviewer would argue otherwise. If this were indeed the case, how can the authors justify their choice of assay conditions (i.e., 1 mM Si over 2 min)? Moreover, in another in-vitro assay typical of Si-transporter studies, time- and dose-dependent curves are routinely conducted.

Response: We previously used *Xenopus* oocyte (*in vivo*) for determining efflux transport activity of Lsi2 in a time- and dose-dependent manner (Nature, 2007).

However, we could not detect the efflux activity of SIET4 when expressed in the oocyte. We therefore changed to employ the proteoliposome assay method (*in vitro*). Compared with oocyte (~1 mm in diameter), the volume of liposomes (~200 nm in diameter) is enormously smaller. Therefore, diffusion between outside and inside the liposome is extremely fast, which results in technical difficulty in assaying the activity in a time dependent manner. However, in response to your comments, we together performed both time- and dose-responsive experiments. We have shown the results in Supplementary Fig. 11. The results support that SIET4 is an efflux transporter for Si.

Re main text:

- Ln 35-36: In line with the comments made by another reviewer, “the absence of Si” is misleading as there are no feasible experimental conditions wherein all Si is removed. A term such as “Si supplementation” (or lack thereof) should be considered (see also Ln 112, 120, etc.)

Response: We have changed as suggested.

- Ln 50: I disagree that “all plants contain significant amount of Si in the above-ground part”. As the authors themselves have noted over the years, plants can be categorized as “passive or rejective” in terms of their Si accumulation – surely these cases would not be considered “significant”.

Response: We have deleted “significant amount” as indicated in the revised manuscript.

- Light language editing still required throughout the ms.

Response: We have asked a language editor for polishing our English.

- Ln 297: This argument requires clarification. If all dicots contain SIET4 (Ln 257), why the “improper” Si deposition in transgenic Arabidopsis?

Response: We have made some discussion on this issue in the revised manuscript. The major problem is that 35S promoter was used.

- Fig 3:

o Caption should specify what the statistical test results relate to for CCCP and Ge treatments.

Response: We have added the caption for the statistical test of these results.

o Again, the data only demonstrate influx, not efflux, contradicting the response letter, and thus failing to support the claim that SIET4 mediates cellular Si efflux.

Response: As we explained above, the results in Fig. 3 show the efflux activity of Si (transport from cytosol (high pH) to apoplast (low pH)).

o This data is further complicated by the fact that the authors described Lsi2 as unidirectional (efflux only) in a previous report (albeit in another heterologous system, i.e., *Xenopus* oocytes; Ma et al., 2007), but demonstrate Lsi2-mediated influx here. This discrepancy should be addressed.

Response: We used Lsi2 as a positive control as suggested last time. As we explained above, both Lsi2 and SIET4 show efflux activity for Si in this assay method. These results are consistent with our previous results on Lsi2 with different assay method (oocyte).

Reviewer #3 (Remarks to the Author):

The revised MS has been improved significantly. However, There are still some points to be addressed.

1) In the Supplementary Figure 6, the shoot biomass of mutant SIET 4-1 grown hydroponically with 0.1, 0.3 or even 1.0 mM Si added was insignificantly different from that of CK, suggesting that the mutant can grow well in solutions with 0.1-1.0 mM Si added like WT. Since the Si concentration in soil solutions normally ranges from 0.1-0.6 mM, we can guess that at a very high concentration of Si, the mutants (SIET 4-1 and SIET 4-2) can survive without significant loss of biomass (especially the shoot biomass) during 21-day growth period. If the authors did similar pot trials with different types of soils containing 0.1-1.0 mM Si in soil solutions with mutants and WT, the mutants would hopefully grow as well as WT. If so, the conclusions as claimed in the title, abstract and conclusion sections cannot be fully supported, i.e. this gene is not essential for survival of rice in all soils. I still believe that there is a threshold or critical value below which the SIET mutant can grow well in soils as WT. We need more evidence using soil culture experiment. From the results of Supplementary Figure 6, it is clear that the two mutants, especially SIET 4-1, can grow well at a very high concentration of Si (0.3-1.0 mM). I suggest that the authors do a supplementary experiment

with soils differing in soil solution Si level (e.g. 0.1, 0.2, 0.5, 1.0 and 2.0 mM) to find out the critical value. This information is critical because it can modify the major conclusion of this MS or make the conclusion more correct or rational.

Response: Thank you for your suggestion! As we showed in Supplementary Fig. 3, the growth of *siet4* mutants gradually decreased with time compared with the wild-type rice, and finally leading to death (Fig. 1). Since we only grew the plants for 21 days, although we observed Si-induced symptoms in the leaves of the mutants even at low Si concentration, the difference in fresh weight was small between the mutants and WT.

In response to your comments, we further performed a sand experiment with different Si application rates under the similar nutrient condition. We used a pure silica gel fertilizer (Water Silica), which is a pure Si fertilizer with slow release. We also monitored the Si concentration in the sand solution. As shown in Supplementary Fig. 6, the Si concentration in the sand solution ranged from less than 0.1 mM to around 1.0 mM, which mimics Si concentration in most soils. The fresh weight of the mutants was decreased with increasing Si application rates. Furthermore, the inhibited growth of the mutants was observed at low Si application rate. Since the Si concentration in solution in most paddy soil is higher than 0.3 mM, the present results strongly support our conclusion.

2) We are still unknown that what is the direct reason to cause plant death for the mutants. I suggest that the authors need to explain in more details in discussion.

Response: We have added some discussion on this issue in the revised manuscript. Since genes related to various stresses were abnormally up-regulated due to improper Si deposition in the mutants, unbalanced growth was probably resulted, finally leading to the death.

3) The English needs polishing by a native speaker because there are still some English errors.

Response: We have asked a language editor for polishing our English.

Reviewer #1 (Remarks to the Author):

All scientific concerns have been adequately addressed. However, the writing in general could still be improved for clarity and grammar. Just a few examples:

Ln 139-40: "...the inhibition of growth was stronger in the *siet4-1* mutants than in WT" – this should be rephrased as it suggests both mutant and WT showed growth inhibition.

Ln 141-2: It is not clear how "improper Si deposition" can be concluded based on these observations.

Ln 197: typo for "Si".

Ln 301-10: This paragraph should be rewritten for clarity.

Ln 328-32: This sentence should be rewritten for clarity.

Ln 377: "...20 µg Si/L without Si added."

Ln 414: "...an aliquot...was mixed..."

Re display items:

Suppl. Fig. 6: The description of panel (a) is unclear – are these values obtained with the WT?

Suppl. Fig. 10: A description of the scale bar is needed.

We thank the reviewer 1 for further comments! We have revised the paper as detailed below.

Reviewer #1 (Remarks to the Author):

All scientific concerns have been adequately addressed. However, the writing in general could still be improved for clarity and grammar. Just a few examples:

Response: Thank you for your positive comments! We have checked the writing throughout the text.

Ln 139-40: "...the inhibition of growth was stronger in the *siet4-1* mutants than in WT" – this should be rephrased as it suggests both mutant and WT showed growth inhibition.

Response: We have rephrased this to "With increasing Water Silica application rates, the growth of WT was enhanced, but that of *siet4-1* mutants was progressively inhibited" in the revised manuscript.

Ln 141-2: It is not clear how "improper Si deposition" can be concluded based on these observations.

Response: We agree that it is too early to make this conclusion based on the result, so we have deleted this sentence.

Ln 197: typo for "Si".

Response: Corrected!

Ln 301-10: This paragraph should be rewritten for clarity.

Response: We have rewritten this paragraph in the revised manuscript.

Ln 328-32: This sentence should be rewritten for clarity.

Response: We have modified this paragraph in the revised manuscript.

Ln 377: "...20 µg Si/L without Si added."

Response: We have corrected this.

Ln 414: "...an aliquot...was mixed..."

Response: We have corrected this.

Re display items:

Suppl. Fig. 6: The description of panel (a) is unclear – are these values obtained with the WT?

Response: To keep the growth condition similar between wild-type rice and mutants, we grew them in the same pot with three independent replicates. We have added this information in the revised figure legend.

Suppl. Fig. 10: A description of the scale bar is needed.

Response: We have added the scale bar in the revised legend.